# Remaining Useful Life Prediction of Rolling Bearings Based on CBAM-CNN-LSTM

**DOI:** 10.3390/s25020554

**Published:** 2025-01-19

**Authors:** Bo Sun, Wenting Hu, Hao Wang, Lei Wang, Chengyang Deng

**Affiliations:** School of Mechanical and Vehicle Engineering, Changchun University, Changchun 130022, China; 220101004@mails.ccu.edu.cn (W.H.); 230102024@mails.ccu.edu.cn (H.W.); 230101019@mails.ccu.edu.cn (L.W.); 220101025@mails.ccu.edu.cn (C.D.)

**Keywords:** convolutional neural network, Convolutional Block Attention Module, deep learning, rolling bearing, remaining service life prediction

## Abstract

Predicting the Remaining Useful Life (RUL) is vital for ensuring the reliability and safety of equipment and components. This study introduces a novel method for predicting RUL that utilizes the Convolutional Block Attention Module (CBAM) to address the problem that Convolutional Neural Networks (CNNs) do not effectively leverage data channel features and spatial features in residual life prediction. Firstly, Fast Fourier Transform (FFT) is applied to convert the data into the frequency domain. The resulting frequency domain data is then used as input to the convolutional neural network for feature extraction; Then, the weights of channel features and spatial features are assigned to the extracted features by CBAM, and the weighted features are then input into the Long Short-Term Memory (LSTM) network to learn temporal features. Finally, the effectiveness of the proposed model is verified using the PHM2012 bearing dataset. Compared to several existing RUL prediction methods, the mean squared error, mean absolute error, and root mean squared error of the proposed method in this paper are reduced by 53%, 16.87%, and 31.68%, respectively, which verifies the superiority of the method. Meanwhile, the experimental results demonstrate that the proposed method achieves good RUL prediction accuracy across various failure modes.

## 1. Introduction

Prognostics and Health Management (PHM) is an integrated technology aimed at enhancing equipment reliability and operational efficiency. One of the key technologies in PHM is to predict the operational status of equipment to minimize losses caused by failures, ultimately to achieve the purpose of predictive maintenance. Rolling bearings serve as critical basic components in mechanical systems, its application covers the automotive industry, rail transportation, agricultural machinery, national defence and military industry and other fields, to maintain the health of mechanical equipment, smooth operation plays a key role. Rolling bearings serve for a long time in extreme working conditions and harsh environments, due to fatigue, wear and tear, corrosion, overload and other factors will lead to performance degradation, which will cause a series of failures, and ultimately lead to safety accidents and property damage [1]. Most of the multiple problems mentioned above are related to the condition of the bearings themselves, which are subject to increasing friction in the course of their work, leading to fatigue, wear and other forms of failure; When there is a decline in the tightness of the bearings or there are process problems in the manufacturing process of the bearings, resulting in the emergence of radial internal clearance, which is also the reason why the bearings will have the above problems; Also, insufficient lubrication problems can lead to corrosion and overloading [2,3,4]. RUL prediction is used to estimate the normal use time of mechanical equipment or its key components in the current operating state, which can not only summarise the operating state of the equipment, but also help to optimise the subsequent management decisions, RUL prediction as the core technology of PHM has the role of the beginning and the end. Therefore, accurate and efficient RUL prediction is of great significance in analyzing bearing degradation trends, optimizing operation and maintenance decisions, reducing equipment failures, and improving equipment utilization, as well as helping to reduce economic losses and avoid accidents [5,6].

Existing principal methods for bearing RUL prediction are primarily classified into two categories [7,8]: RUL prediction approaches driven by model and RUL prediction approaches. Model-driven RUL prediction methods primarily rely on mathematical and physical models to describe the failure and degradation mechanisms of systems or equipment, so as to realize RUL prediction.Model-driven RUL-based prediction methods can still provide high-precision prediction results when the model is constructed stably and the parameters are determined reasonably well, but the lack of an in-depth understanding of the internal mechanism of the system will lead to a model that is not accurate enough or difficult to be applied in practice.Data-driven RUL-based prediction methods, on the other hand, rely on degraded data, do not require in-depth knowledge of the physical or engineering principles of the system, and can automatically extract features and learn them with the help of machine learning, which avoids the problem of degraded modelling and reduces human intervention and subjectivity, and is therefore increasingly being used [9].

Data-driven based approaches are utilizing historical degradation data to train network models for RUL prediction.In recent years, with the innovation of sensor technology and the development of big data processing technology, Deep learning-based methods for RUL prediction have increasingly emerged as a prominent area of research. Ren et al. [2] developed a new feature extraction technique leveraging deep convolutional neural networks to extract degenerate features and used them for RUL prediction, and achieved good prediction accuracy. Wang et al. [10] utilized a convolutional autoencoder to construct a Health Indicator (HI) as a label for the bearing vibration data, and then input the labeled data into a one-dimensional CNN network, which ultimately realized the bearing RUL prediction. Cao et al. [11] introduced a novel deep learning model that integrates the residual attention mechanism with temporal convolutional networks, and proved through experimental validation that the model has better performance in long-term prediction tasks in RUL prediction. Huang [12] utilized a deep learning approach focusing on the spatio-temporal features of the data and introduced A global attention mechanism for weighting features, and finally used a multi-branch CNN for RUL prediction. Xie et al. [13] used expanded CNN to extract multi-scale spatio-temporal features from normalized signals, and then used CAM for weighting features, and finally achieved the prediction of bearing RUL. Marco et al. [14] utilized instantaneous spectral entropy (ISE) and continuous wavelet transform (CWT) to monitor the health status of rotating machinery, and provided important theoretical support for the unsubsequent RUL prediction.

Most of the current studies use a single attention mechanism. Channel attention only emphasizes the significance of various channels, while spatial attention only highlights the significance of different spatial locations, if a single attention mechanism is used it will ignore the important features in the other dimension, resulting in incomplete feature selection. The Convolutional Block Attention Module (CBAM) is an attention mechanism designed to improve Convolutional Neural Network (CNN) performance. CBAM improves feature representation by adding an attention module behind the convolutional block, which enables the model to adaptively focus on important information in the input feature map. He et al. [15] proposed a bearing RUL prediction model based on multiscale CNN, feature extraction is performed by multiscale CNN, and CBAM is embedded in the CNN structure to realize the allocation of the importance of different features, and finally the RUL prediction of bearings is realized on the bearing dataset. Wang et al. [16] proposed a bearing RUL prediction model based on CBAM and temporal convolutional network, where temporal convolutional network is used for feature extraction and CBAM is used for adaptive weight assignment of different features, and finally the model is verified to have good generalization on the public dataset. Yu et al. [17] first added the improved Inception V1 module to the CNN structure to enhance the CNN’s capability for comprehensive feature extraction, then added the CBAM module to the CNN structure to weight the features extracted by the CNN, and experimentally confirmed the proposed model’s validity. Zhang [18] and others used continuous wavelet transform (CWT) to transform the data dimensions, and then used CBAM and ResNet to construct the bearing RUL prediction model, and finally successfully realized the bearing RUL prediction under two working conditions. Xiang et al. [19] proposed a dynamic residual network (Dy-ResNet) combined with CBAM for bearing RUL prediction on the basis of ResNet network, and the experiments proved that Dy-ResNet is much better than ResNet prediction.

To this end, this paper proposes a residual life prediction method based on CBAM, which first uses FFT to perform data preprocessing on the original vibration signal data of rolling bearings, and inputs the processed signal data into CNN to extract features, aiming at extracting the deep features in the signal data that contain rich degradation information. Then CBAM is used to assign attention weights to the features, focusing on the key feature information and ignoring the influence of unimportant features. The channel attention mechanism in CBAM can enhance the expression of channel feature information, while the spatial attention mechanism focuses on the representation of important spatial feature information in local areas, which are complementary and mutually advantageous.Finally, the long-term dependency of the time-series data is learned through LSTM network, which enables the model to perform RUL prediction more accurately. The proposed model makes up for the lack of close connection of channel and spatial features in the field of bearing RUL prediction.The CBAM attention mechanism module is used in combination with the CNN-LSTM network model and has shown excellent performance in a wide range of application scenarios such as in image analysis and processing, fault diagnosis, life expectancy prediction, and medical disease diagnosis.Compared to the RUL prediction models proposed in literature [20] and literature [21], the CBAM-CNN-LSTM model has better generalization and better prediction performance. Comparing to the literature [20], where the models in that literature were evaluated using MSE for prediction performance, the MSE of the model in this paper is reduced by 12.28% and 10.34%. Comparing with the literature [21], which uses MSE, MAE, and RMSE for prediction performance evaluation, the model in this paper has 53%, 9.75%, and 31.68% reduction in the three metrics, respectively.

The paper is organized as follows: Section 2 provides the theoretical background. Section 3 gives the network structure of the bearing RUL prediction method and the process of bearing RUL prediction. Section 4 performs experimental validation to verify the feasibility and effectiveness of the proposed model on a publicly available bearing dataset. Conclusion of the proposed method is given in Section 5.

## 2. Theoretical Background

### 2.1. Convolutional Neural Network

CNN are feed-forward neural network inspired by biological vision systems and have been broadly utilized in computer vision techniques and natural language processing, equipment fault diagnosis and health management in recent years [22]. CNN is mainly composed of five parts: input layer, output layer, convolutional layer, pooling layer, and fully connected layer, and its core components are convolutional layer, pooling layer and fully connected layer. The structure diagram of CNN network is shown in Figure 1.

For the convolutional layer, the most important thing is the convolutional kernel, which performs local region sliding operation on the input degenerate data with the size of the convolutional kernel window, and the result of the convolutional computation is the feature figure composed of local features.The CNN convolutional layer possesses the outstanding advantage of weight sharing, which reduces the difficulty of model complexity and also avoids overfitting caused by too many network parameters [23].

Assuming that the vibrational data input to the convolutional layer of the CNN model is *X*, then the output of the convolutional layer is calculated as follows:(1)$$C_{n}=\sigma \left(W_{n}⊗X+b_{n}\right)$$
where: $C_{n}$ is the nth hidden layer feature of the convolution output; $W_{n}$ denotes as the weight matrix; $b_{n}$ denotes as the bias matrix; σ(·) denotes as the activation function; ⊗ denotes the convolution operation; *n* is the number of convolution kernels.

Pooling layer is also known as downsampling or subsampling, the main role of the pooling layer is to reduce the feature dimensions of the output information of the convolutional layer, to reduce the amount of computation on the basis of retaining the important information, and to improve the speed of the network operation. The main pooling functions are maximum pooling function, average pooling function, global average pooling function, global maximum pooling function [24]. The maximum pooling function is the most widely used pooling function. The formula is as follows:(2)$$P_{n}=maxC_{n}$$
where: $P_{n}$ denotes as the result after pooling; $C_{n}$ denotes as the input before pooling.

The fully-connected layer, sometimes called the densely-connected layer, is usually found at the end of the CNN and is essentially a nonlinear transformation and linear combination of deep features extracted from the convolutional and pooling layers [25]. The role of the fully connected layer is to fuse and connect the deep features obtained from the previous neural network layers and use the final one-dimensional features as the output values for the target task.

### 2.2. Convolutional Attention Mechanism

The attention mechanism is an information allocation mechanism that mimics the human visual mindset and works by focusing core attention on crucial information while reducing the interference of redundant information as a way to improve model performance, among other things. Currently, a variety of attention mechanisms combined with neural network algorithms have been applied to different tasks and have achieved good results. CBAM integrates Channel Attention Mechanism (CAM) and Spatial Attention Mechanism (SAM) into a hybrid attention framework. The advantage of CBAM is that while focusing on the spatial hierarchy of channel features, it can also focus on the local characterization information [26], which is filtered by weight allocation to filter out two dimensions of important degradation feature information to enhance the contribution to the RUL prediction task.

Since the degree of contribution of different output channels of the convolutional layer to the RUL prediction task is inconsistent, the importance of different output channels is assigned weights using the channel attention mechanism, and more important output features are selected to enhance the impact of important channel features on the RUL prediction task.

The channel attention mechanism is illustrated in Figure 2, where the input feature dimension is S × C, where S is the spatial dimension and C is the channel dimension. First of all, the channel attention mechanism uses both Global average pooling (GAP) and Global Maximum Pooling (GMP) to pool the degraded features; then the degraded features with dimensions of 1 × C are input into the contributing Multi-Layer Perceptron (MLP) to perform the feature summation and fusion; and then the Sigmoid function is used to perform the normalization operation on the obtained results, and the degraded feature channel attention weight is obtained in the end [27]. The formula for the channel attention mechanism is:(3)$$M_{c}\left(F\right)=\delta \left[MLP\left(GAP\left(F\right)\right)+MLP\left(GMP\left(F\right)\right)\right]$$
where: *F* is the input features; $M_{c}\left(F\right)$ is the corresponding channel weight and δ(·) is the Sigmoid activation function.

The spatial attention mechanism focuses on the location of key information on the spatial dimension of the input features and enhances the impact of important features on the spatial dimension on the RUL prediction task. The principle is shown in Figure 3.

Similarly, the spatial attention mechanism performs pooling operations using both GMP and GAP on the input feature channel dimension to obtain a feature vector with both dimensions H × 1; then the two feature vectors are spliced along the direction of the channel dimension to form a characteristic vector of H × 2, and then the feature vector of H × 1 is regained through the convolutional layer operation; and finally, the degraded feature is obtained by using the Sigmoid function for the normalization process Spatial Attention Weights. The spatial attention mechanism is formulated as:(4)$$M_{s}\left(F\right)=\delta \left(Conv\left(\left(GMP\left(F\right);GAP\left(F\right)\right)\right)\right)$$
where: $M_{s}\left(F\right)$ is the corresponding spatial attention weight; Conv is the single kernel convolution.

As a hybrid attention mechanism, CBAM complements the strengths of both CAM and SAM by combining them for use, but different combinations of the two can also have differences in effectiveness.There are 3 combinations of CBAM, which are SAM followed by CAM in series; CAM followed by SAM in series; and CAM and SAM in parallel. Due to the fact that SAM is first used for local key feature weighting, there exists inadequate processing of local feature information, which will affect the effectiveness of the subsequent CAM in weighting the channel features. In addition, the parallel connection of CAM and SAM amplifies the randomness of the CBAM network, making the CBAM network less stable. In view of the above reasons, the CBAM network combination adopted in this paper is a tandem approach using CAM first and SAM later. CAM is used first for important channel weighting, and then SAM is used to enhance the contribution of local critical features and suppress the expression of non-critical features [28]. The network structure of CBAM is shown in Figure 4.

The overall CBAM formula is as follows:(5)$$F_{1}=M_{c}\left(F\right)⊗F$$(6)$$F_{2}=M_{S}\left(F_{1}\right)⊗F_{1}$$

The calculation process of the channel attention mechanism is shown in Equation (Equation 10): the input feature *F* is weighted by the channel attention weights to obtain $F_{1}$.

The calculation process of the spatial attention mechanism is shown in Equation (Equation 11): for the weighted feature $F_{1}$ is then weighted by the spatial attention mechanism to get the final weighted feature $F_{2}$.

### 2.3. Long Short-Term Memory Network

LSTM is a distinct variant of Recurrent Neural Network (RNN), which was co-proposed by two AI experts in 1997, LSTM as a variant of RNN aims to solve the problems of gradient explosion, gradient vanishing, and difficulty in dealing with long-term dependencies that can occur in the training of RNN [29]. Specifically, LSTM introduces a gating mechanism consisting of input gates, output gates, and forgetting gates to control the input, output, update, and forgetting of information, and the introduction of the gating mechanism gives LSTM an obvious advantage in capturing long-term dependencies in temporal data [30]. The structure of the LSTM network is illustrated in Figure 5.

The forgetting gate $f_{t}$ of LSTM can control to discard those information. Its formula is:(7)$$f_{t}=\sigma \left(W_{f}\cdot \left[h_{t-1},x_{t}\right]+b_{f}\right)$$
where: $x_{t}$ is the current input; $h_{t-1}$ is the output of the previous moment; $W_{f}$ is the weight matrix of the forgetting gate; $b_{f}$ is the bias of the forgetting gate; σ(·) is the Sigmoid activation function.

The input gate $i_{t}$ of the LSTM controls the flow and update of information [31], which is formulated as:(8)$$i_{t}=\sigma \left(W_{i}\cdot \left[h_{t-1},x_{t}\right]+b_{i}\right)$$(9)$$\tilde{C_{t}}=tanh\left(W_{c}\times \left[h_{t-1},x_{t}\right]+b_{c}\right)$$(10)$$C_{t}=f_{t}\times C_{t-1}+i_{t}\times \tilde{C_{t}}$$
where: $W_{i}$ is the weight matrix of the input gate; $W_{c}$ is the weight matrix of the current cell state; $b_{i}$ is the bias of the input gate; $b_{c}$ is the bias of the cell state; $C_{t-1}$ is the cell state at the previous moment; $C_{t}$ is the cell state at the current moment; $\tilde{C_{t}}$ is the candidate cell state; tanh is the hyperbolic tangent function.

The LSTM output gate $o_{t}$ can control the output of the message, which is calculated as:(11)$$o_{t}=\sigma \left(W_{o}\cdot \left[h_{t-1},x_{t}\right]+b_{o}\right)$$
where: $W_{o}$ is the weight matrix of the output gate; $b_{o}$ is the bias of the output gate.

## 3. Bearing RUL Prediction Methods and Processes

### 3.1. Network Model Structure

The RUL prediction method proposed in this paper combines the advantages of CNN, LSTM, and CBAM, and gives full play to their advantages on the basis of rational arrangement of the structure of each module, and focuses the used resources on improving the RUL accuracy and generalization of the target task. The network model structure of the method is shown in Figure 6.

Three convolutional layers of CNN are used for feature extraction of the FFT-processed frequency-domain amplitude signals, while two maximum pooling layers are used for feature dimensionality reduction in order to reduce the model computation and improve the model RUL prediction accuracy.

The valid features extracted using CNN are used as inputs to the CBAM module inserted in front of the third maximum pooling layer, which successively uses CAM and SAM to select valid information for the features, and subsequently assigns weights to the information of varying importance.

The LSTM layer captures the long-term dependencies contained in the significant features after weighted assignment. Finally, local feature fusion is performed through a single fully-connected layer to complete the label mapping from degraded features to RUL and output a predicted value.

In order to improve the network training speed and mitigate overfitting, a batch normalization layer is introduced between the model convolution layer and and the maximum pooling layer. Batch normalization reduces the sensitivity of the model to data distribution and increases the generalization ability of the model. The batch normalization for batch input data $B=[x_{1},x_{2}\cdots x_{m}]$ is calculated as:(12)$$\mu_{b}=\frac{1}{m}\sum_{i=1}^{m}x_{i}$$(13)$$\sigma_{b}^{2}=\frac{1}{m}\sum_{i=1}^{m}{\left(x_{i}-\mu_{b}\right)}^{2}$$(14)$${\hat{x}}_{i}=\frac{x_{i}-\mu_{b}}{\sqrt{\sigma_{b}^{2}}+\epsilon }$$(15)$$y_{i}=\gamma {\hat{x}}_{i}+\beta$$
where: $\mu_{b}$ is the mean of the batch input data; $\sigma_{b}^{2}$ is the variance of the batch input data; ${\hat{x}}_{i}$ is the normalized input; $y_{i}$ is the output after scale transformation and bias; γ and β are the parameters that need to be learned when the network is trained.

### 3.2. Rul Prediction Process

Figure 7 illustrates the RUL prediction flowchart for the proposed method and the specific steps of the process are given below:

Step 1: RUL prediction was performed using the publicly available dataset PHM2012. The acquired raw vibration signal data of rolling bearings are subjected to Fast Fourier Transform to get the frequency domain amplitude signal for the purpose of data preprocessing. And the obtained dataset is divided into training set and test set.

Step 2: The RUL prediction used in this paper is an “end-to-end” prediction method, which is a RUL prediction method that directly inputs data from the input end to the output end to get the prediction result, and defines the range of RUL label values as (0, 1). The CBAM-CNN-LSTM network model is constructed using CNN, LSTM, and CBAM modules, and the ECA-CNN-LSTM network model is constructed to compare with the model proposed in this paper.

Step 3: The network model of the proposed method is trained using the training set of bearings under three different operating conditions and the model parameters are adjusted simultaneously to achieve the best RUL prediction. The model hyperparameters are tuned using Adam optimizer and the forward propagation algorithm is used to adjust the network model weights to arrive at a loss function value that minimizes the error between the predicted and true values.

Step 4: The trained model is tested using the selected bearing test set and the prediction results are obtained, the model prediction effect is evaluated using performance evaluation metrics, and a model comparison is made with the comparison model to conclude the superiority and effectiveness of the model in this paper.

## 4. Experimental Validation

### 4.1. Introduction to the Dataset

In this paper, we use the experimental dataset from the PHM Data Challenge held in 2012, which was acquired by the PRONOSTIA test rig under different loading conditions. The PRONOSTIA test rig is shown in Figure 8. The vibration signals were acquired from the horizontal direction and the vertical direction of the PRONOSTIA test rig, respectively [32]. During the signal acquisition process, the data acquisition was carried out every 10 s with a sampling frequency of 25.6 kHz and a sampling time of 0.1 s. The PHM2012 dataset was initially used in the RUL Prediction Challenge, the association organizing the competition has set up three working conditions the first two bearings as the training set and the rest as the test set, the training set is characterized by a small amount of data and does not give the type of faults that occurred, which makes it difficult to perform an accurate RUL. At the same time, this dataset includes multiple bearing data under various operating conditions, the bearing data is large in scale, the data is a complete full-life degradation cycle data, and the data is of high quality, so we chose the PHM2012 dataset for the prediction model training.

The test dataset contains the full life cycle degradation data of 17 rolling bearings in three different working conditions, seven sets of data for bearings 1-1 to 1-7 in working condition one, seven sets of data for bearings 2-1 to 2-7 in working condition two, and three sets of data for bearings 3-1 to 3-3 in working condition three. The test uses six sets of bearing data across varying operating conditions as the training set, that is, bearings 1-1, 1-2, 2-1, 2-1, 3-1, 3-2, and bearings 1-3, 1-4, 1-5, 1-7, 2-3, 2-6 as the test set. Table 1 below shows the test data sets for the bearings.

Rolling bearings in the operation of the process there are two different failure modes, namely, the performance of gradual degradation of the failure mode, the performance of sudden failure mode. Performance gradual degradation failure mode is characterized by the bearing vibration amplitude over time will continue to grow, but its overall degradation trend is relatively slow; performance sudden failure mode is characterized by the bearing in the early part of the operating state is relatively stable, but the later part of the amplitude of its vibration will be drastically changed. According to Figure 9a,b bearing 1-3 is a typical performance degradation failure mode, and bearing 2-6 is a typical performance sudden failure mode. In order to visualize the two failure modes of rolling bearings, the proposed method can simultaneously predict the RUL of rolling bearings under two failure modes, bearing 1-3 and bearing 2-6 are selected from the test set for experimental analysis.

### 4.2. Data Preprocessing

Original vibration signals from rolling bearings contain extensive performance degradation information, but the larger data dimension will affect the subsequent RUL prediction work, and data preprocessing of bearing data is needed. And FFT is widely used in the field of data signal processing and other fields, and can efficiently complete the processing task [33]. Therefore, FFT was performed on 12 rolling bearings for 3 conditions to convert the time domain magnitude signals into frequency domain magnitude signals. Since the data collected in the horizontal direction of the test dataset contains more valid information, the subsequent tests are trained and tested using vibration data in the horizontal direction. Taking the sampling data in the horizontal direction within the first time of 0.1 s for bearing 1-1 as an example, the original time-domain amplitude signal is shown in Figure 10a, and the frequency-domain amplitude signal after FFT is shown in Figure 10b.

### 4.3. Evaluation Indicators

In order to accurately assess the validity of the performance of the proposed RUL prediction model, MSE, RMSE, MAE, and coefficient of determination $R^{2}$ are often used as model performance evaluation indexes. MSE, RMSE and MAE, as classical evaluation indexes, the closer their values are to 0, the better the predictive ability of the model and the higher the prediction accuracy. And both of them measure the size of the deviation between the true value and the predicted value, but MSE, RMSE is more sensitive to the outliers, which can reflect the performance of the model to the extreme values, and MAE focuses on the overall error level of the model, which can reflect the average prediction accuracy of the model. However, MSE, RMSE and MAE alone may not effectively indicate the quality of the model, introducing $R^{2}$ is essential to measure the fit between predicted and true values, $R^{2}$ advantage lies in the model for the interpretation of the data, $R^{2}$ value range is [0,1], A higher $R^{2}$ value, closer to 1, indicates better model fit, with predicted values more closely aligning with the true values.

In the face of complex models or different application scenarios, a single performance evaluation metric cannot help us fully understand the performance of a model. Therefore, by choosing three performance evaluation metrics, namely, MSERMSE, MAE, and $R^{2}$, to evaluate a prediction model, this allows for a more thorough and precise evaluation of model performance and ensures the model’s effectiveness and robustness across various real-world applications.

The *MSE*, *RMSE*, *MAE*, and $R^{2}$ formulas are:(16)$$MSE=\frac{1}{N}\sum_{i=1}^{N}{\left(y_{i}-{\hat{y}}_{i}\right)}^{2}$$(17)$$RMSE=\sqrt{\frac{1}{N}\sum_{i=1}^{N}{\left(y_{i}-{\hat{y}}_{i}\right)}^{2}}$$(18)$$MAE=\frac{1}{N}\sum_{i=1}^{N}\left(\left|y_{i}-{\hat{y}}_{i}\right|\right)$$(19)$$R^{2}=1-\frac{\sum_{i=1}^{N}{\left(y_{i}-{\hat{y}}_{i}\right)}^{2}}{\sum_{i=1}^{N}{\left(\bar{y_{i}}-y_{i}\right)}^{2}}$$
where: *N* represents the number of bearing samples, $y_{i}$, ${\hat{y}}_{i}$, $\bar{y_{i}}$ is the actual value, predicted value, and mean value corresponding to the first sample.

### 4.4. Parameter Setting

Aside from the CBAM-CNN-LSTM model presented in this paper, CNN-LSTM, and an ECA-CNN-LSTM model that combines a channel attention mechanism (ECA) and CNN-LSTM are also selected as comparison models. Table 2 below shows the main parameters of the CBAM-CNN-LSTM model.

In addition, all models utilize MSE as the loss function, with the Adam optimizer and a learning rate of 0.001, a batch size of 128, and a number of iterations of 100. The experimental environments are torch1.13 and Python3.4, and the models are trained using only the CPU, which is model i5-8265U (Intel Corporation, CA, USA), and the RAM is 8 GB.

When we set the size of the convolution kernel, the first layer kernel size is set to be larger than 32 in order to avoid missing feature information, and then the kernel size of the next two layers is set to be smaller, which is also in line with the principle of CNN layer-by-layer convolution for extracting important local features, and therefore the kernel size of the last two layers is set to be 10. Such a size setup can improve the robustness and prediction performance of the model. The selection of the step size is the same as the kernel size selection, the larger step size 2 is selected first for data dimensionality reduction, and the subsequent step size is set to 1 aiming at retaining the local feature information, Secondly, the kernel size and step size should match the input data, so no arbitrary changes can be made. For the setting of the number of CNN convolutional layers, multi-layer convolutional operation can indeed extract deeper feature information, but will inevitably bring problems such as overfitting, therefore, choosing the appropriate number of 3-layer convolutional layers not only improves the model prediction performance, but also prevents overfitting and reduces the waste of computational resources. In order to verify our theory, bearings 1-3, the performance evaluation index $R^{2}$, were selected for the sensitivity analysis of the number of convolutional layers of the CNN in Table 2, When the convolutional layer is 3 to 6 layers, $R^{2}$ is 0.943, 0.924, 0.908, 0.9431 respectively, and the experimental study proved that the selected parameters have been After repeated experimental comparisons and integrating the model prediction effect and the time used, the optimal parameters were selected.

### 4.5. Analysis of the Test Results

Comparing the proposed method with the introduced CNN-LSTM and ECA-CNN-LSTM models, it can be seen from Table 3 that the prediction performance of the CNN-LSTM model without CBAM decreases more significantly, which proves that the CBAM module, which can be assigned to different channels and spatial feature weights, can make the model pay more attention to the important information, and has a significant effect on the improvement of the model RUL prediction performance has a significant effect. In contrast to the ECA-CNN-LSTM model for single-dimensional features, the CBAM-CNN-LSTM model, because the CBAM module is a mixture of two attention mechanisms in series, allows the model to focus on the spatial structure of the features while also focusing on the information of the local features, which enables the CBAM-CNN-LSTM model to capture degraded data in a more comprehensive way. degradation information in both dimensions, adapting to complex and diverse scenarios.

In this paper, three performance evaluation metrics, MAE, RMSE, and $R^{2}$, are employed to evaluate model performance. Table 3 lists the three performance comparison metrics of MSE, MAE, RMSE, and $R^{2}$ for three different models for the six bearings in the test set. The MSE, MAE and RMSE values of the CBAM-CNN-LSTM model for test bearings 1-3 are the smallest among the three models, which are 0.047, 0.037 and 0.069, respectively. Compared to the MSE, MAE and RMSE values of the ECA-CNN-LSTM model, reductions of 76.85%, 67.54% and 51.74% are observed. The MSE, MAE value and RMSE value of the comparative CNN-LSTM model are reduced by 14.73%, 66.36% and 43.90%; its $R^{2}$ value is the highest, 0.943, and the $R^{2}$ value of the comparative ECA-CNN-LSTM model and CNN-LSTM model is improved by 19.8% and 13.4%. From Table 3, it can still be concluded that the MAE, MSE, and RMSE values of the test bearing 2-6 model CBAM-CNN-LSTM are still lower than those of the other comparison models, and its $R^{2}$ value remains superior to that of the comparison models.

In addition, in order to verify the superiority of the proposed method in an all-round way, 100 experiments are carried out and averaged for the proposed model and the comparison model, and the experiments show that, although the proposed method is more complicated than the comparison method, the consumed time of 19 min is only slightly higher than that of the two comparison methods of 18 and 16 min, which meets the requirements of real-time prediction and practicability, and the comparison models of ECA-CNN-LSTM and CNN-LSTM, although consuming less time, the prediction effect is far inferior to that of the proposed method. Table 3 illustrates that the constructed model is not only applicable to bearing RUL prediction for different failure modes, but also capable of more accurate RUL prediction.

Figure 11 and Figure 12 show the prediction effects of the test bearings on the three models. Figure 11 illustrates, the predicted RUL values of bearings 1-3 fluctuate up and down in accordance with the true values, and the error with the true values is small, which aligns with the gradual performance degradation typical of the failure mode, and the CBAM-CNN-LSTM model exhibits higher overall prediction accuracy than the other two models. From Figure 12, the predicted RUL value for bearing 2-6 shows minimal deviation from the actual value, even with fewer data points, and the first half of it fits the real value, and the second half of it deviates from the real value due to its performance of the sudden failure modes; however, the CBAM-CNN-LSTM model consistently outperforms the other two models in prediction performance across all stages.

The network model proposed in this paper is applicable to the RUL prediction of rolling bearings under different degradation modes and can achieve good prediction results. For rolling bearings in the performance degradation failure mode, the degradation process is relatively smooth, so for this degradation mode, the impact on the RUL prediction model belongs to the conventional category, we use CNN, LSTM in the design of the network model can be very good to capture the smooth degradation trend in the data. As for the rolling bearings in the performance sudden failure mode, the degradation process is more complicated, and there are drastic changes in the late degradation stage, compared with the performance gradual degradation mode, there is only a certain amount of degradation information in the features extracted in this stage, so the prediction of the bearing RUL in this mode is more difficult. For such degradation modes, we use a hybrid attention mechanism CBAM module on top of the CNN-LSTM network, where the CAM and SAM in CBAM can be weighted for key features, i.e., the important degradation information present in the emergent phase is weighted with attention. How to carry out the network model design, both can realize the bearing RUL prediction in the two degradation modes.

In order to reflect the superiority of the proposed method, the prediction results of the network model proposed in this paper are compared with the literature [20] and the literature [21] using the same dataset and the same training set of bearings, and the results are shown in Table 4, taking bearings 1-3 and bearings 2-6 as examples.Literature [20] uses only MSE for performance evaluation, while literature [21] uses MSE, MAE and RMSE as performance evaluation indexes for prediction models. The comparison shows that the prediction errors of this paper are lower than literature [20] and literature [21], and the prediction errors of other test bearings selected in this paper are lower than literature [20]. For literature [21], except for bearings 1-7 and bearings 2-3, the prediction errors of the other bearings are lower than that of literature [21]. Overall, the method proposed in this paper is better than the literature [20,21].

## 5. Conclusions

This paper presents a method for predicting bearing RUL combining convolutional attention mechanism and CNN-LSTM, and draw the following conclusions:

FFT does the data preprocessing to prepare for subsequent feature extraction. CNN performs the feature extraction, and the extracted features are subjected to CBAM for the allocation of attention weights, while LSTM learns the temporal features of the features, ultimately enhancing the RUL prediction accuracy of the bearings and realizes RUL prediction under different failure modes.

The prediction results show that the MSE, MAE and RMSE of this study are reduced by 53%, 9.75%, and 31.68%, respectively, in comparison with the existing bearing RUL prediction methods, which proves the effectiveness and superiority of the proposed method.

The data-driven ToMFIR technique can integrate multi-source information technology, and in future research, this technique can be effectively combined with existing neural network models, which will further improve the performance of RUL prediction for rolling bearings.

## Figures and Tables

**Figure 1 sensors-25-00554-f001:**
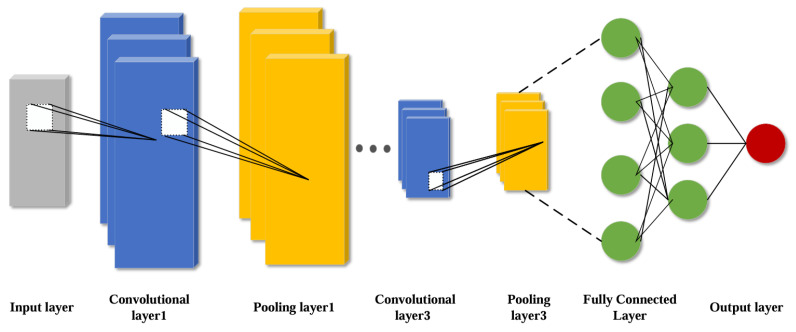
Schematic diagram of CNN structure.

**Figure 2 sensors-25-00554-f002:**
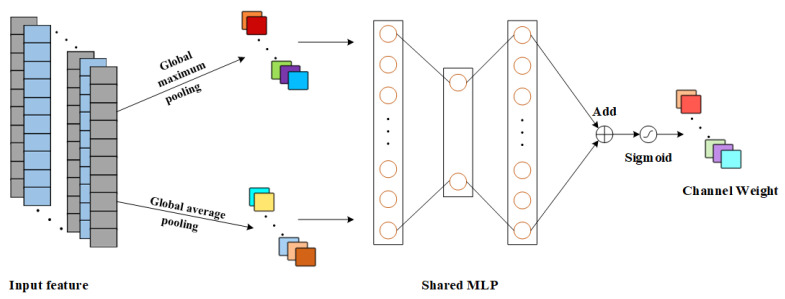
Schematic diagram of CAM principle.

**Figure 3 sensors-25-00554-f003:**
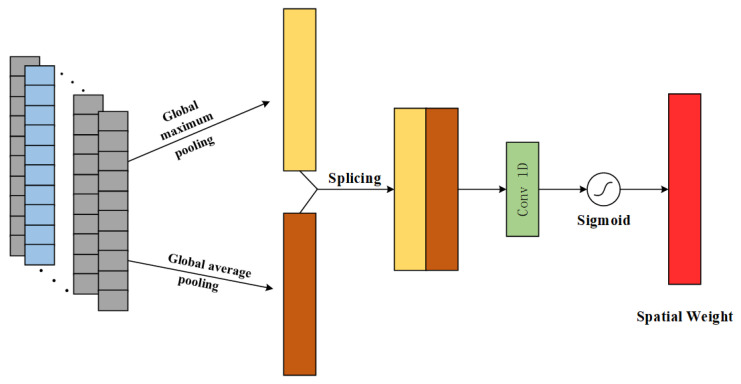
Schematic diagram of SAM principle.

**Figure 4 sensors-25-00554-f004:**
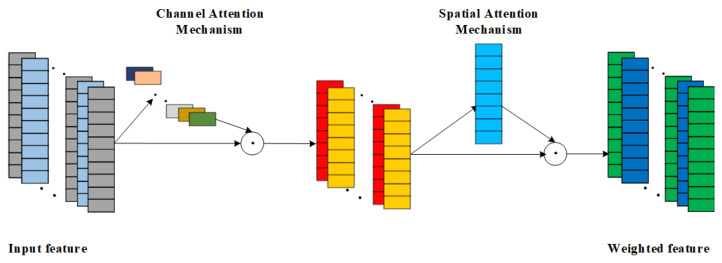
Schematic diagram of CBAM structure.

**Figure 5 sensors-25-00554-f005:**
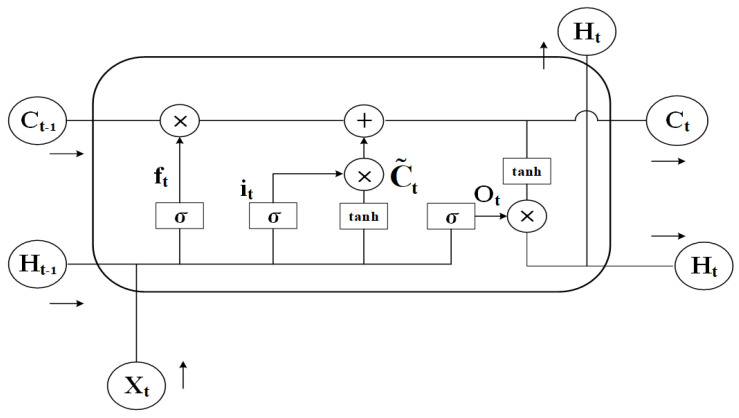
Schematic diagram of LSTM structure.

**Figure 6 sensors-25-00554-f006:**
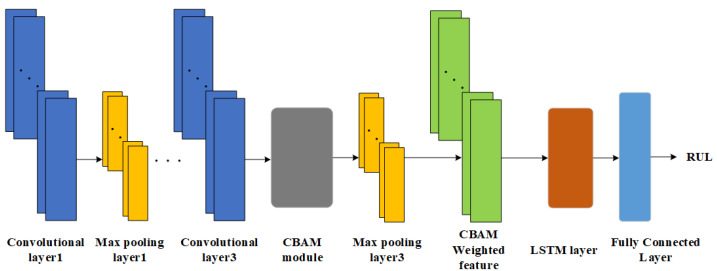
Schematic diagram of CBAM-CNN-LSTM structure.

**Figure 7 sensors-25-00554-f007:**
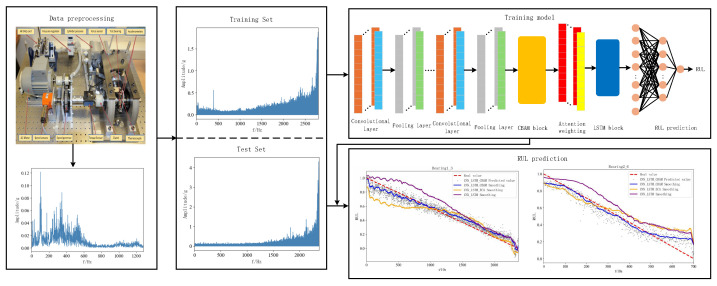
Flowchart of RUL prediction process.

**Figure 8 sensors-25-00554-f008:**
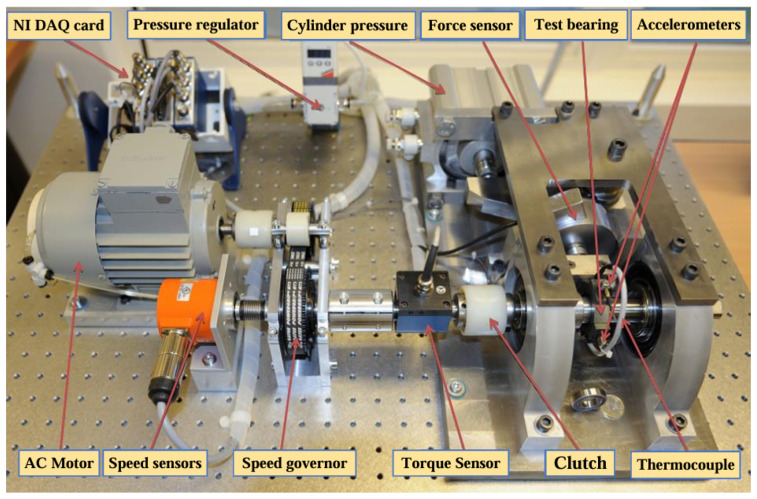
PRONOSTIA test bench.

**Figure 9 sensors-25-00554-f009:**
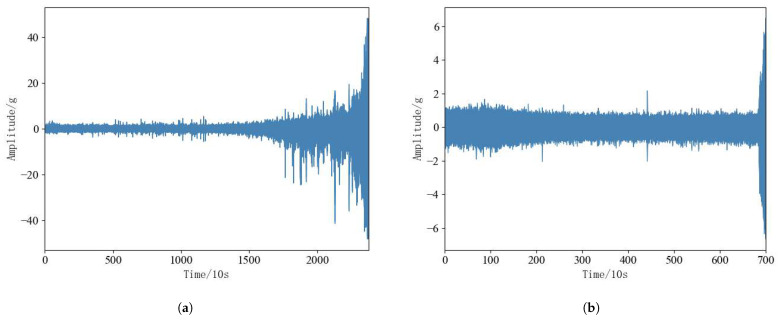
(**a**): Bearing 1-3. (**b**): Bearing 2-3.

**Figure 10 sensors-25-00554-f010:**
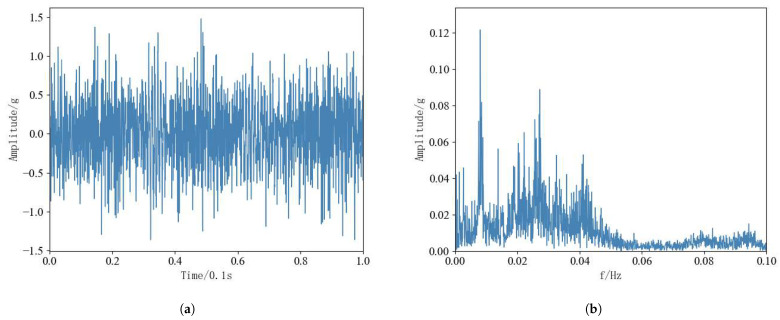
(**a**): Bearing 1-1 Time-domain signal in 0.1 s. (**b**): Bearing 1-1 Frequency domain signal in 0.1 s.

**Figure 11 sensors-25-00554-f011:**
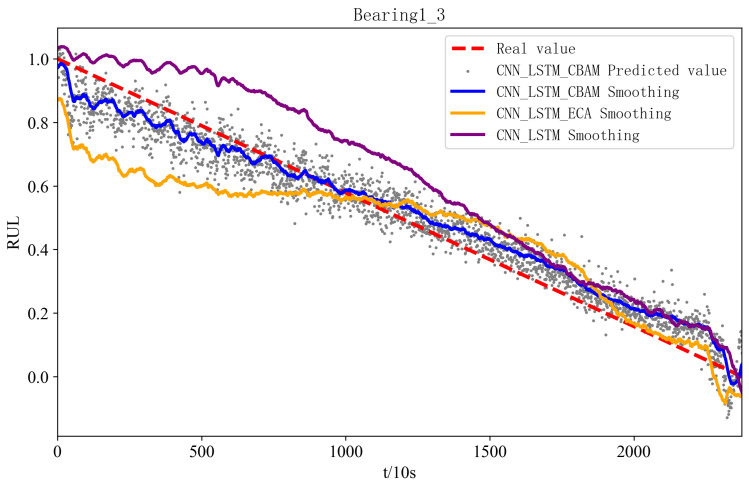
RUL figure for bearing 1-3 with different models.

**Figure 12 sensors-25-00554-f012:**
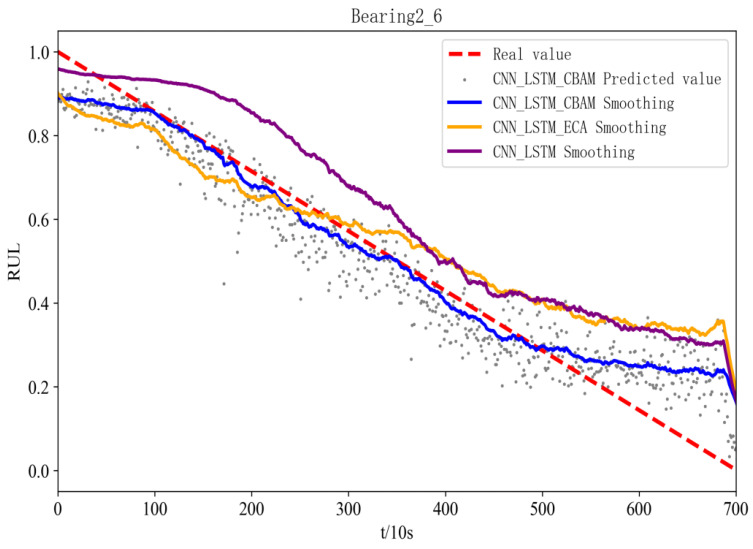
RUL figure for bearing 2-6 with different models.

**Table 1 sensors-25-00554-t001:** Description of the bearing dataset.

	Operating Condition		
	Condition 1	Condition 2	Condition 3
Load (N)	4000	4200	5000
Rotate speed (r/min)	1800	1650	1500
	1-1	2-1	3-1
	1-2	2-2	3-2
	1-3	2-3	
Bearing dataset	1-4	2-4	
	1-5	2-5	3-3
	1-6	2-6	
	1-7	2-7	

**Table 2 sensors-25-00554-t002:** Parameter configuration of the CBAM-CNN-LSTM.

Network Layer	Parameters
Conv 1	channels = 1, filters = 16, kernel size = 32, Stride = 2, Activation function = ReLu
Conv 2	channels = 16, filters = 32, kernel size = 10, Stride = 1, Activation function = ReLu
Conv 3	channels = 32, filters = 64, kernel size = 10, Stride = 1, Activation function = ReLu
LSTM layer	Hidden size = 128, Num layers = 2
CBAM block	kernel size = 7, Reduction ratio = 16

**Table 3 sensors-25-00554-t003:** Performance comparison of different methods on test set.

Indicator	Model Name	Bearing 1-3	Bearing 1-4	Bearing 1-5	Bearing 1-7	Bearing 2-3	Bearing 2-6
MSE	CBAM-CNN-LSTM	0.0047	0.0077	0.0198	0.0137	0.0223	0.0085
	ECA-CNN-LSTM	0.0203	0.0183	0.0258	0.0154	0.0284	0.0163
	CNN-LSTM	0.0152	0.0176	0.0256	0.0337	0.0296	0.0151
RMSE	CBAM-CNN-LSTM	0.069	0.086	0.141	0.117	0.148	0.088
	ECA-CNN-LSTM	0.143	0.132	0.155	0.121	0.167	0.134
	CNN-LSTM	0.123	0.139	0.146	0.181	0.174	0.128
MAE	CBAM-CNN-LSTM	0.037	0.060	0.091	0.078	0.112	0.069
	ECA-CNN-LSTM	0.114	0.103	0.115	0.085	0.126	0.106
	CNN-LSTM	0.110	0.099	0.090	0.135	0.139	0.105
	CBAM-CNN-LSTM	0.943	0.910	0.762	0.836	0.737	0.906
R^2^	ECA-CNN-LSTM	0.756	0.790	0.710	0.825	0.664	0.783
	CNN-LSTM	0.817	0.765	0.744	0.605	0.636	0.803

**Table 4 sensors-25-00554-t004:** Description of the bearing dataset.

	Literature [20]	Literature [21]			Proposed Method		
Test Bearing	MSE	MSE	MAE	RMSE	MSE	MAE	RMSE
Bearing 1-3	0.0054	0.01	0.041	0.101	0.0047	0.037	0.069
Bearing 2-6	0.0089	0.042	0.083	0.205	0.0078	0.069	0.088

## Data Availability

Public datasets used in our paper: https://github.com/wkzs111/phm-ieee-2012-data-challenge-dataset (accessed on 16 January 2025).

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
