# Peer review of "Remaining Useful Life Prediction of Rolling Bearings Based on CBAM-CNN-LSTM"

_sensors, 2025, doi:10.3390/s25020554_

Round 1
Reviewer 1 Report
Comments and Suggestions for Authors
This manuscript presents a novel method for predicting RUL by combining CBAM and CNN-LSTM. However, the following comments need to be addressed before it is considered for acceptance.
1. This manuscript is poor in English, should be significantly improved for its English grammar, sentence structure and content flow. Careful revision of manuscript is necessary.
2. On page 1, ABSTRACT, the authors need to compare the proposed method with existing methods to demonstrate the value of the proposed method.
3. On page 3, INTRODUCTION, the advantages and application context of the proposed method should be elaborated in detail.
4. The tables in article should follow the standard format.
5. The horizontal coordinates in Figure 9-12 should begin at 0. All figures should be checked carefully and prepared as per guidelines of the journal.
6. On page 9, Section 3.2, the authors need to provide more comprehensive and detailed elaboration on RUL prediction process.
7. Figure 7 should be further optimized as it is too rough.
8. The presentation of experimental evaluation is limited to Table 3 and Figure 13-14, which need be expanded appropriately.
9. On page 15, CONCLUSIONS, should be updated with more critical observations and conclusions.
Comments on the Quality of English Language
This manuscript is poor in English, should be significantly improved for its English grammar, sentence structure and content flow. Careful revision of manuscript is necessary.
Reviewer 2 Report
Comments and Suggestions for Authors
This paper presents a method for predicting bearing RUL combining CBAM and CNN-LSTM, Overall, the paper is well written and organized with a proper length. The given results seem correct. However the following issues should be clearly addressed in the revision:
1. The innovation of this paper is not clear and it is difficult for readers to understand the main contributions of this paper , especially in comparison with the large number of existing references on bearing life prediction based on neural networks. This part should be added in Introduction section.
2. The description of the existing work should be shorter in Introduction section. Furthermore, more descriptions of the proposed method are needed.
3. How do the two fault modes mentioned in the article (gradual degradation failure mode and sudden failure mode) specifically affect the design and training of RUL prediction models? How can these two fault modes be effectively distinguished in the model?
4. Incipient fault is an important research issue, and its existing work is widely applied on fault diagnosis and remaining useful life prediction. The reviewer recommends that the author incorporate recent advancements in 'Data-Driven ToMFIR' technique as a reference for future work and cite relevant articles, given that the integration of analytical models with data-driven approaches is an inevitable trend in the aforementioned research directions.
Reviewer 3 Report
Comments and Suggestions for Authors
The research paper
‘Remaining Useful Life Prediction of Rolling Bearings Based on Convolutional Block Attention Module
By Bo Sun et al,
focuses on a deep learning approach for predicting the Remaining Useful Life (RUL) of rolling bearings
The authors propose a hybrid model that integrates a Convolutional Block Attention Module (CBAM) with a combined Convolutional Neural Network and Long Short-Term Memory network (CNN-LSTM). This CBAM-CNN-LSTM architecture is then applied for feature extraction from raw vibration signals.
Overall, the paper is of interest to researchers in civil and building engineering, especially for the ones focusing on the evergrowing field of Structural Health Monitoring (SHM).
Nevertheless, several issues, related to both the paper’s content and format, are enlisted here below and should be addressed in order to achieve full acceptance.
1. The title of the paper is too similar to “Remaining useful life prediction of rolling bearings based on convolutional recurrent attention network” by Qiang Zhang et al (http://dx.doi.org/10.1108/AA-08-2021-0113)
2. The iThenticate report highlights a much higher-than-usual similarity with already-published papers (>30%).
3. The PHM2012 bearing dataset was used; however, the rationale for selecting this dataset over others should be explicitly stated. Additionally, details on how this dataset ensures the robustness and generalizability of the proposed method could be clarified.
4. Furthermore, this is called PHM2012 in the abstract and PRONOSTIA test bench in Section 4. Are these two names for the same dataset? or is it a typo?
5. The authors compared their proposed CBAM-CNN-LSTM model against CNN-LSTM and ECA-CNN-LSTM models. Expanding this comparison to include other state-of-the-art RUL prediction methods could strengthen the validation of the proposed model.
6. While the network parameters are provided, the process for selecting these parameters (e.g., kernel size, step size, number of layers) and their potential impact on model performance could be elaborated upon. A proper sensitivity analysis is needed for this kinds of ML applications.
7. Why is the procedure evaluated only in terms of RMSE, MAE, and R²? other measures such as precision, recall, or F1-score (especially in scenarios of critical RUL thresholds) might provide a more comprehensive performance evaluation.
8. the paper should include a more thorough review of relevant literature, to position the proposed method within the context of existing research. Especially, the Authors should more clearly report in detail any research gaps that the proposed method aims to fill. Related to this point, the Authors may add the following recent works on bearing fault diagnosis: An Application of Instantaneous Spectral Entropy for the Condition Monitoring of Wind Turbines and Non-destructive techniques for the condition and structural health monitoring of wind turbines: A literature review of the last 20 years.
9. The paper lacks a detailed discussion of computational efficiency. Including benchmarks for training and prediction times relative to model complexity would help readers assess practical usability.
10. There are repetitive sentences in the introduction and other sections (e.g., on the importance of RUL prediction and generalizations about attention mechanisms). The paper should be better double-checked and reviewed for English and formatting.
Reviewer 4 Report
Comments and Suggestions for Authors
I found your article very interesting titled “Remaining useful life prediction of rolling bearings based on convolutional block attention module”, but in my opinion below remarks would improve your manuscript under the scientific level.
Comments and Suggestions for Authors:
1. In the Abstract I suggest to add the main results of the prediction accuracy of the proposed methodology.
2. Lines 22-25 you refer to the fatigue coming from the different issues. In this place I suggest to find the proper references regarding this issue. Moreover, I suggest to refer the fatigue to the operational parameters of rolling-element bearings such as friction torque, radial internal clearance or amount of lubrication. I suggest to add also 3 following papers to your list of references:
· Measuring micro-friction torque in MEMS gas bearings, Sensors 2016, 16(5), 726,
· Intelligent diagnostics of radial internal clearance in ball bearings with machine learning methods, Sensors 2023, 23(13), 5875,
· Oil-Air distribution prediction inside ball bearing with under-race lubrication based on numerical simulation, Applied Sciences 2024, 14(9), 3770.
3. I’m not sure about the Section 2, while the methods are described in details, I think it is the best to refer to the own analysis and specifying own input parameters to the neural networks. My suggestion is to shorten this part and put it in the place of analysis.
4. Figure 7, I wouldn’t say that it is a flowchart, it is just the illustration. Nevertheless, please put the detailed description what is going to be presented in that figure, that it is an essential figure for the topic, so the text after the Figure should be placed before it.
5. Section 4, as you are referring to the FFT analysis, the type of the bearing must be specified. The characteristic frequencies are referring to the internal design of it and you should provide the list of characteristic frequencies of it.
6. There is something wrong with the data presented in Figure 9 and 10. You have mentioned that the sampling frequency is 25.6kHz, what is really high value for such tests, how did you get the failure of bearing after around 7 hours (Figure 9). Could you present it in hours maybe, I don’t believe that bearing can get failure after short time. I think that test should consider the load capacity of bearing and the nominal value of time should be taken as reference. Nevertheless, more details must be specified on the test rig and type of bearing.
7. Figure 13 and 14 are hard to comprehend. Please also provide the results in form of the Table or in form of confusion matrix.
8. Conclusions are really plain and the accuracy of the method must be specified, otherwise, the manuscript shouldn’t be recommended for its publishing.
Round 2
Reviewer 2 Report
Comments and Suggestions for Authors
In a general way most of my comments were answered by the authors. My overall opinion about this paper is quite good. The manuscript is well written and acceptable for publishing
Reviewer 3 Report
Comments and Suggestions for Authors
After reviewing the authors' responses to the reviewers' comments, it is this Reviewer’s opinion that most of the replies are satisfactory. The explanations appear comprehensive and clear. No explicit avoidance or refusal to modify suggestions was noted. The authors made an effort to address each comment
Therefore, the paper can be tentativelly accepted, with minor corrections: please try to fix some formatting issues, such as the lack of journal logo in the forefront of the first page, and to increase the quality of the pictures, which seem to be blurred.
Also, please provide next time a highlighted modified version, to make the double-check easier for the Reviewers
Reviewer 4 Report
Comments and Suggestions for Authors
Almost all remarks have been introduced to the revised version of the manuscript. I don't see the reference to the question regarding what type of bearing have been used and there is no information about FFT results. I suggest to fulfill this information, and in revised version there are no highlights. Conditionally, I recommend the manuscript for its publishing, but other reviewers should refer also to my remarks.